

# Differential transcriptome analysis and identification of genes related to resistance to blight in three varieties of *Bambusa pervariabilis* × *Dendrocalamopsis grandis*

Fengying Luo[1,*], Xinmei Fang[1,*], Han Liu[2], Tianhui Zhu[1], Shan Han[1], Qi Peng[1] and Shujiang Li[1,3]

[1] College of Forestry, Sichuan Agricultural University, Chengdu, Sichuan Province, China
[2] Ganzi Institute of Forestry Research, Kangding, Sichuan Province, China
[3] National Forestry and Grassland Administration Key Laboratory of Forest Resources Conservation and Ecological Safety on the Upper Reaches of the Yangtze River, Chengdu, Sichuan Province, China
[*] These authors contributed equally to this work.

Corresponding author
Shujiang Li, 14087@sicau.edu.cn

## ABSTRACT

**Background.** *Bambusa pervariabilis* × *Dendrocalamopsis grandis* is a fast-growing bamboo that is widely introduced in southern China and has great economic and ecological benefits. In recent years, a blight of *B. pervariabilis* × *D. grandis* caused by *Arthrinium phaeospermum* has led to much branch damage and even death of entire bamboo forests.

**Methods.** To screen for resistance genes in *B. pervariabilis* × *D. grandis*, transcriptome sequencing technology was used to compare the gene expression profiles of different varieties of *B. pervariabilis* × *D. grandis* with variable resistance and the same varieties under different treatments. The Clusters of Orthologous Groups of Proteins (COG) database; the Gene Ontology (GO) database; and the Kyoto Encyclopedia of Genes and Genomes (KEGG) database were used to annotate and analyse the differentially expressed genes.

**Results.** A total of 26,157 and 11,648 differentially expressed genes were obtained in the different varieties after inoculation with *A. phaeospermum* and the same varieties after inoculation *A. phaeospermum* or sterile water, respectively. There were 23 co-upregulated DGEs and 143 co-downregulated DEGs in #3 and #8, #6 and #8, #6 and #3. There were 50 co-upregulated DGEs and 24 co-downregulated DEGs in the same varieties after inoculation *A. phaeospermum* or sterile water. The results showed that many genes involved in cell wall composition synthesis, redox reactions and signal transduction were significantly different after pathogen infection. Twenty-one candidate genes for blight resistance, such as *pme53*, *cad5*, *pod*, *gdsl-ll* and *Myb4l*, were found. The qRT-PCR results were consistent with the sequencing results, verifying their authenticity. These results provide a foundation for the further exploration of resistance genes and their functions.

## INTRODUCTION

*Bambusa pervariabilis* × *Dendrocalamopsis grandis* is the product of the female parent *B. pervariabilis* and the male parent *D. grandis*. After 12 years of hybrid cultivation, four excellent hybrid bamboo varieties, #3, #6, #8 and #30, were identified. *B. pervariabilis* × *D. grandis* is widely used in papermaking and returning farmland to forest, which also promotes the construction of ecological barriers in the middle and lower reaches of the Yangtze River. However, in recent years, it has been found that hybrid bamboo shoot blight caused by *Arthrinium phaeospermum* (*Arthrinium phaeospermum* (Corda) M.B. Ellis) infection has become an important disease in the cultivation area of hybrid bamboo (*Ma et al., 2003*). After the *B. pervariabilis* ×*D. grandis* is susceptible to the disease, the brown spots first appear near the bamboo nodes, and then the spots gradually expand. When the disease spots surround the bamboo nodes, the branches above the disease spots turn yellow and the bamboo leaves fall off, and the whole plant dies in serious cases. It was found that the shoot blight of hybrid bamboo caused damage to more than 3,000 hm$^2$ of hybrid bamboo, causing large economic losses (*Zhu et al., 2009*) and greatly threatening the process of returning farmland to forest and the construction of ecological barriers in the Yangtze River Basin.

The resistance to plant pathogens varies among plant-resistant and plant-susceptible varieties, which is important for the analysis of pathogenic mechanisms and the utilization of disease-resistant resources. At present, the research in this field in China and abroad mainly focuses on the following aspects: selecting representative varieties to compare and analysing the cell histological changes of different resistant and susceptible varieties after infection (*Pereira et al., 2013*; *Carisse et al., 2000*; *Vallad & Subbarao, 2008*), studying enzymes related to disease resistance (*Lin et al., 2014*), and revealing the related mechanism of resistant and susceptible varieties through omics technology (*Collinge et al., 2008*). In particular, the development of omics technology in recent years has guided further exploration of the interaction between plants and pathogens. The resistance of *B. pervariabilis* × *D. grandis* was induced by pre-inoculation of toxin protein, and then the gene expression before and after induction was compared by transcriptome sequencing. It was found that redox proteins, phenylalanine ammonia lyase and S-adenosine-L-methionine synthase were highly expressed. These results may indicate that these genes are related to the disease resistance level of *B. pervariabilis* ×*D. grandis*. (*Peng et al., 2020*). *Li et al. (2019)* Separated and identified the proteins before and after *B. pervariabilis* × *D. grandis* infected with *A. phaeospermum* by TMT (tandem mass tag)-labeled quantitative protein technology combined with LC-MS/MS (mass spectrometry), and screened the related proteins that may be involved in *B. pervariabilis* × *D. grandis* disease resistance, including cinnamyl alcohol dehydrogenase, a protein homologous to maize cysteineprotease 1 and other proteins. After infection with *Candidatus* Liberibacter asiaticus, there were differences in the expression of genes related to transcription factors, secondary metabolism, receptor kinase and hormone signal transduction in the resistant and susceptible varieties of grapefruit (*Wang et al., 2016*). Through subsequent verification, two genes related to the resistance of *C. Liberibacter asiaticus*, *DMR6-like* and *NPR1-like*, were predicted.

The transcriptome refers to the collection of all transcription products in cells at a certain developmental stage or under certain physiological conditions and can be used to study gene expression, gene function, and structure. In recent years, this technology has been widely used to screen plant disease-resistance-related substances and identify key differentially expressed genes (*Liu et al., 2014*; *Li et al., 2014*; *Salgado et al., 2014*). Researchers used transcriptome sequencing technology to sequence susceptible and resistant tomatoes after inoculation with *Cladosporium fulvum* and verified the important role of the *Cf* gene in tomato resistance to leaf mould caused by *C. fulvum* (*Zhang et al., 2020*). Guijiao 9 and Williams are resistant and susceptible to *Fusarium oxysporum*, respectively. After the transcriptome data of Guijiao 9 and Williams infected with *F. oxysporum* were analysed, it was found that genes related to defence-related metabolite synthesis, such as Nb-LRR protein, calmodulin-binding protein and phenylpropanoid biosynthesis genes, were significantly upregulated in resistant varieties, which confirmed the important role of plant hormone regulation and defence-related gene activation in banana disease resistance (*Sun et al., 2019*).

Past forest surveys showed that there were differences in the disease index among different resistant varieties of *B. pervariabilis* × *D. grandis*. Therefore, by comparing the transcriptome data from different varieties, differentially expressed genes related to disease resistance in *B. pervariabilis* × *D. grandis* could be found, which is the key to disease resistance breeding and biological control of *B. pervariabilis* × *D. grandis* shoot blight. In this study, transcriptome sequencing technology was used to analyse the transcriptome of different varieties of *B. pervariabilis* × *D. grandis* under different treatment conditions. The gene expression profiles of *B. pervariabilis* × *D. grandis* between different resistant and susceptible varieties and the gene expression profiles of the same varieties under different treatments were compared to obtain candidate genes for shoot blight resistance. To lay the foundation for further exploration of resistance genes, the functional genes identified here are important genetic resources to improve the resistance of *B. pervariabilis* × *D. grandis* to shoot blight based on gene technology.

# MATERIALS AND METHODS

## Plant materials and pathogens

*A. phaeospermum* was stored at the Laboratory of Plant Pathology, Sichuan Agricultural University, and one-year-old *B. pervariabilis* × *D. grandis* from varieties #3 (moderately resistant), #6 (resistant), and #8 (susceptible) (*Li et al., 2018*) were planted in the greenhouse near the fifth teaching building of Chengdu Campus of Sichuan Agricultural University.

## Inoculation and disease investigation

The preserved hyphae were inoculated on PDA medium, and then inoculated on sugarcane juice medium (sugarcane juice 30–100 g/L, magnesium sulfate heptahydrate 1.0 g/l, potassium dihydrogen phosphate 1.0 g/L, agar 20 g/L) and cultured under the dark condition of 25 °C for 10 days. The spores were washed with sterile water, and the conidial suspension was diluted to $10^6$ cfu/ml. The 30 one-year-old hybrid bamboos of varieties #3, #6 and #8 were inoculated, with three biological replicates. Five plants of the three varieties

inoculated with sterile water were used as controls. The incidence rate and disease index were calculated after 25 days of inoculation.

$$I(\%) = D/T \times 100$$

where I, D and T are the incidence, the number of diseased plants and the number of total plants, respectively.

The disease classification standard was as follows: 0: no wilt; Level 1: less than 25% withered branches; Level 2: 25%–50% (including 25% and 50%) withered branches; Level 3: 50%–75% withered branches (including 75%); and Level 4: more than 75% withered branches (*Fang, 1998*).

$$DI = \Sigma(DN \times L)/(TN \times M) \times 100$$

where DI is the disease index, DN is the disease level, L is the number of plants at each level, TN is the total number of diseased plants, and M is the maximum disease incidence.

## RNA extraction and high-throughput sequencing

RNA was extracted from the shoots of hybrid bamboos (#3, #6 and #8) inoculated with sterile water or pathogens for 25 days. The #3, #6 and #8 *B. pervariabilis × D. grandis* treatment groups inoculated with *A. phaeospermum* were named J3, J6 and J8 respectively, and the #3, #6 and #8 *B. pervariabilis × D. grandis* treatment groups inoculated with sterile water were named J3, J6 and J8 respectively. The concentration and purity of the total RNA were determined by an Agilent 2100 analyser (Agilent, Santa Clara, CA, USA). We constructed RNA-Seq libraries following the Illumina protocol for mRNA-sequencing sample preparation. The cDNA library was sequenced using an Illumina HiSeq2000 platform by Shanghai Baipu Biotechnology Co., Ltd. After processing the original data as previously described in *Peng et al. (2020)*, Specifically hisat2 (*Kim, Langmead & Salzberg, 2015*) was used to map reads to the reference genome (*Peng et al., 2013*) (https://bioinformatics.psb.ugent.be/plaza/versions/plaza_v4_monocots/organism/view/Phyllostachys+edulis#collapsePanel) and reference transcriptome (GenBank accession in NCBI: txid38705). Genomic sequences and information of *Bambusa pervariabilis × Dendrocalamopsis grandis* were deposited in the NCBI database (BioProject accession number: PRJNA545783, BioSample accession number: SAMN11928018, TSA accession number: GJJA00000000, accessible with the following link: https://www.ncbi.nlm.nih.gov/bioproject/545783).

## Differentially expressed gene analysis

Gene expression calculations were performed according to the FPKM method (*Peng et al., 2020*). The software DEGseq (*Wang et al., 2010*) was used to conduct differential expression analysis among samples. A false discovery rate (FDR) $\leq$ 0.05 and fold change (FC) >2 were used as the standards to identify the differentially expressed genes (DEGs) among different treatments. DEGs annotation were collected as previously described in Peng (*Peng et al., 2020*). Specifically, the DEGs were annotated by NCBI's nonredundant protein sequence database (NR) (*Deng et al., 2006*), the Clusters of

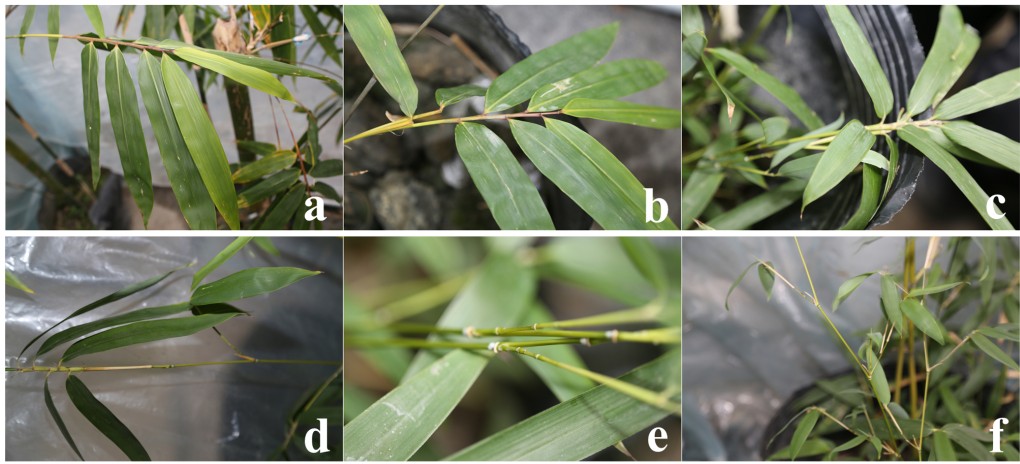

**Figure 1** **Symptom map of *B. pervariabilis* ×*D. grandis* before inoculation.** (A, B, C) #3, #6 and #8 *B. pervariabilis* × *D. grandis* stems, respectively. (D, E, F) #3, #6 and #8 *B. pervariabilis* × *D. grandis* leaves, respectively.

Orthologous Groups of Proteins (COG) database (*Tatusov, Galperin & Natale, 2000*), the Gene Ontology (GO) database (*Ashburner et al., 2000*), and the Kyoto Encyclopedia of Genes and Genomes (KEGG) database (*Kanehisa et al., 2004*). Each database reference link is https://blast.ncbi.nlm.nih.gov/Blast.cgi, http://www.ncbi.nlm.nih.gov/COG/, http://www.geneontology.org/, http://www.genome.jp/kegg/.

### Real-time PCR validation

Primers (Table S1) were designed for five randomly selected genes and 21 disease resistance candidate genes. Primer Premier 5.0 (Premier Biosoft International, Palo Alto, CA, USA) was used for qRT-PCR primer design. The primers were synthesized by Tsingke Biotechnology Co., Ltd. GAPDH was used as an internal reference gene. TransScript® Green One-Step qRT-PCR SuperMix (trans, China) was used for qRT-PCR. The q-PCR system consisted of the following: 10 uL qPCR SuperMix, 0.4 uL RT/RI Enzyme Mix, 0.4 uL Passive Reference Dye, 7.4 uL ddH2O, 0.4 uL F/R, 1 uL RNA. The qPCR was performed: at 45 °C for 5 min, 94 °C for 30 s, 94 °C for 5 s, and 60 °C for 30 s. The third step and the fourth step were repeated 40 times, and a dissolution curve was added at the final step. Each treatment group underwent qPCR three times, and the average value was calculated.

## RESULTS

### Statistics of potted disease experiments performed in the greenhouse

*B. pervariabilis* × *D. grandis* of three different resistant varieties grew well before inoculation with pathogens, and there were no symptoms such as withered branches and leaves (Fig. 1). 25 days after inoculation with the pathogen, #6 the leaves of the high disease resistant variety were slightly chlorotic, #3 the middle resistant variety had brown spots at the bamboo joints, some leaves withered and turned yellow, #8 the susceptible variety had brown spots at the bamboo joints, the branches gradually withered, most of the leaves turned yellow and fell

**Table 1  Statistical table of the disease index of the three varieties.**

| Species | Incidence (%) | Disease index |
|---|---|---|
| #8 | $85.6 \pm 1.7^{a}$ | $62.8 \pm 2.1^{a}$ |
| #3 | $23.3 \pm 3.3^{b}$ | $12.2 \pm 1.7^{b}$ |
| #6 | $7.8 \pm 1.7^{c}$ | $2.8 \pm 0.4^{c}$ |

**Notes.**

Data are the means of three repetitions; data in the same column followed by different lowercase letters indicate significant differences among different species after inoculation by the LSD test ($p < 0.05$)

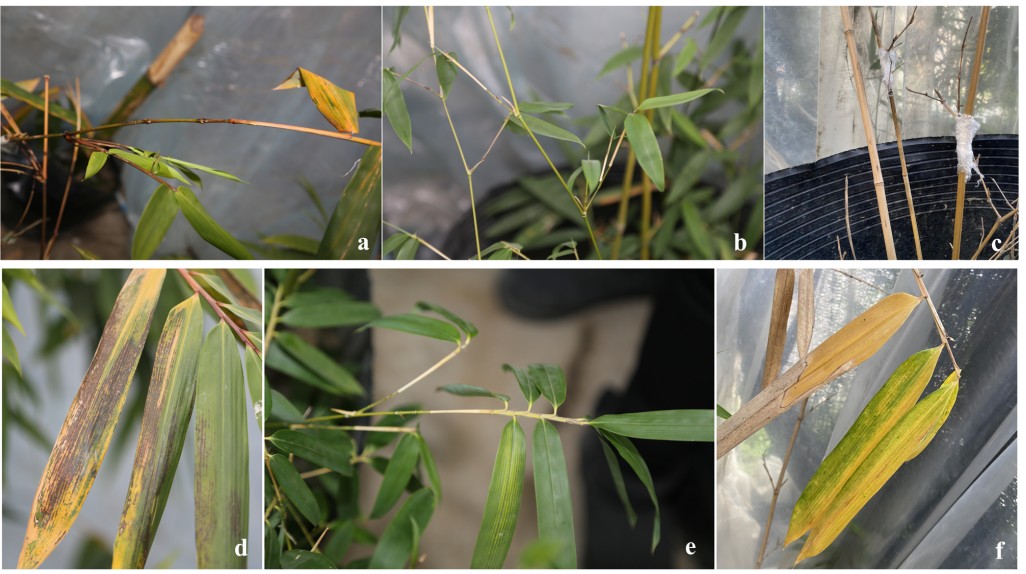

**Figure 2  Symptom map of *B. pervariabilis* × *D. grandis* after inoculation.** (A, B, C) #3, #6 and #8 *B. pervariabilis* × *D. grandis* stems were inoculated for 25 days, respectively. (D, E, F) #3, #6 and #8 *B. pervariabilis* × *D. grandis* leaves were inoculated for 25 days, respectively.

off, and some diseased plants finally died. The incidence and disease index of #3, #6 and #8 hybrid bamboos inoculated with *A. phaeospermum* were statistically analysed (Table 1). The results showed that the incidence was 23.3%, 7.8% and 85.6% and the disease index was 12.2, 2.8 and 62.8, respectively, for hybrid bamboos #3, #6 and #8. According to statistical analysis, #3, #6, and #8 hybrid bamboo inoculated with *A. phaeospermum* had significantly different incidences and disease indices (Fig. 2). According to the resistance standard for shoot blight (*Li, 2013*), hybrid bamboos #3, #6, and #8 showed moderate resistance, full resistance and susceptibility to shoot blight caused by *A. phaeospermum*.

## RNA sequence analysis and annotation results between *A. phaeospermum* infection or sterile water inoculation in the same varieties

Comparative analysis of transcriptomic expression data was determined between *A. phaeospermum* infection and sterile water inoculation in the same varieties. The results showed that 4,387, 6,316 and 4,159 DEGs were obtained in the #3, #6 and #8 varieties,

**Table 2** **Statistical table of the number of DEGs.** J indicates inoculated *A. phaeospermum*, S indicates inoculated sterile water.

| Comparison | Number of DEGs | Upregulated | Downregulated |
|---|---|---|---|
| J3-vs-S3 | 4387 | 1925 | 2462 |
| J6-vs-S6 | 6316 | 3148 | 3168 |
| J8-vs-S8 | 4159 | 2609 | 1550 |
| J6-vs-J3 | 18401 | 4856 | 13545 |
| J6-vs-J8 | 18496 | 5222 | 13276 |
| J3-vs-J8 | 4070 | 2380 | 1690 |

respectively. Among the 4,387 DEGs, 1925 were upregulated and 2,462 were downregulated in variety #3 after inoculation. Among the, 6316 DEGs found for variety #6 after inoculation, 3148 were upregulated and 3,168 were downregulated. Of the 4,159 DEGs found for variety #8 after inoculation, 2,609 were upregulated and 1,550 were downregulated (Table 2).

In total, 394 DEGs were shared across the three varieties (Fig. 3); of them, 50 DEGs were upregulated in the three varieties, and 24 DEGs were downregulated (Table S2).

In the COG database, 1,513, 2,776, and 1,421 DEGs were found for in the same varieties (including #3, #6 and #8) after inoculation with *A. phaeospermum* or sterile water, respectively. In the comparisons between #3, #6, and #8 inoculated *A. phaeospermum* or sterile water, genes were annotated to 25 functional classifications in the GO database (Fig. S1), including "translation", "ribosomal structure and biogenesis", "translation", "replication, recombination and repair", "posttranslational modification, protein turnover, chaperones", "signal transduction mechanisms" and others. In #6 *B. pervariabilis* × *D. grandis* inoculated with *A. phaeospermum* and sterile water, 333, 333 and 316 DEGs were annotated to the classifications "signal transduction mechanisms", "translation" and "replication, recombination and repair" in the COG database (Fig. 4).

In the GO database, 3,506, 5,674 and 3,414 DEGs of #3, #6 and #8 *B. pervariabilis* × *D. grandis* were annotated to the cell component, molecular function and biological process categories after inoculation with *A. phaeospermum* or sterile water, respectively (Fig. S2). For #6 hybrid bamboo inoculated with pathogens, the DEGs were mainly enriched in "metabolic process", "cellular process", "single-organism process" and "response to stimulus" in the biological process classification. In the cell component classification, they were mainly enriched in "cell", "cell part", "organelle" and "membrane", and in the molecular function classification, they were enriched in "binding", "catalytic activity", "transporter activity" and "nucleic acid binding transcription factor activity" (Fig. 5).

In the KEGG enrichment analysis of DEGs in #3, #6 and #8 *B. pervariabilis* × *D. grandis* inoculated with *A. phaeospermum* or sterile water, 791, 1,505 and 782 DEGs were annotated to 121, 125 and 118 metabolic pathways, respectively (Fig. S3). In the KEGG enrichment analysis of the 791 DEGs from hybrid bamboo #3 inoculated with pathogens, most DEGs were enriched in "phenylpropanoid biosynthesis". In the KEGG enrichment analysis of the 1505 DEGs from hybrid bamboo #6 inoculated with pathogens, most DEGs were enriched in "carbon metabolism", followed by "phenylpropanoid biosynthesis" and "starch and sucrose metabolism". In the enrichment analysis of the 782 DEGs from hybrid
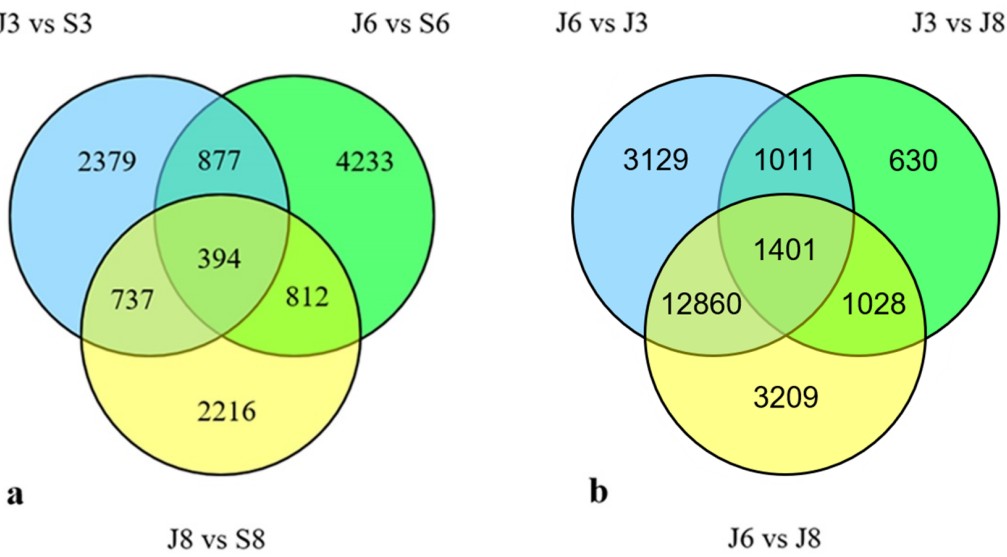

**Figure 3 Venn diagram of DEGs.** (A) Venn diagram showing the DEGs between the sterile water and spore suspension treatments for varieties #3, #6 and #8. (B) Venn diagram showing the DEGs between varieties #3, #6 and #8 after spore suspension treatment. (J indicates inoculated *A. phaeospermum*, S indicates inoculated sterile water).

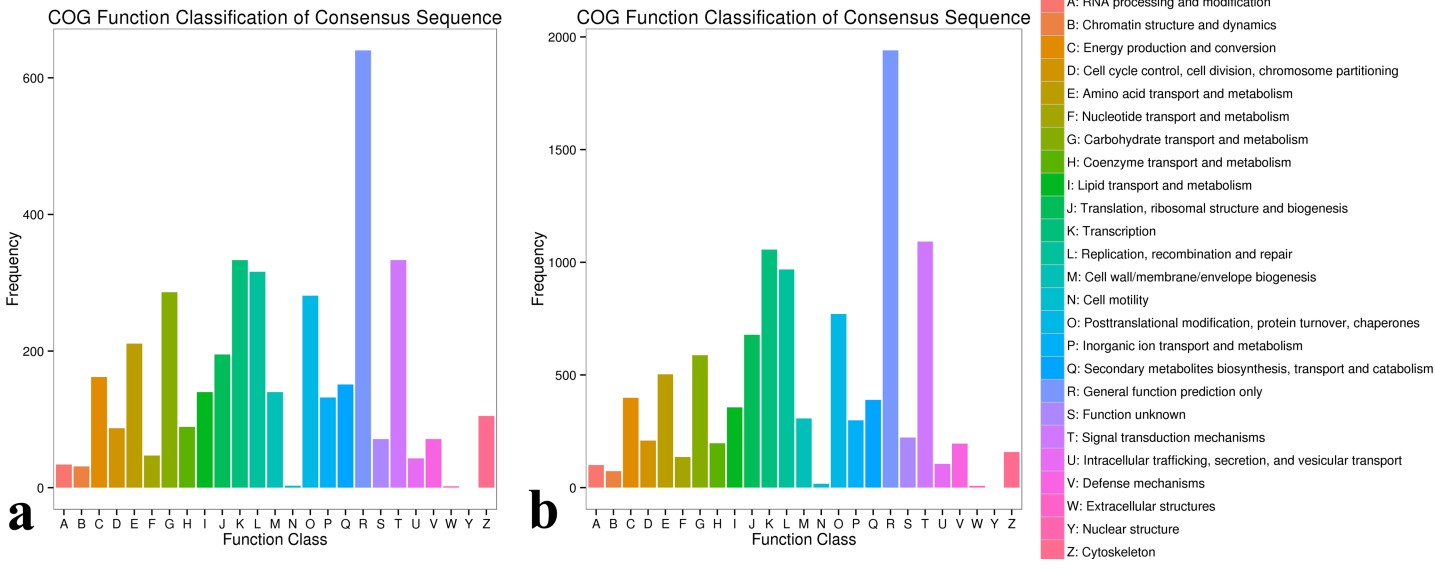

**Figure 4 COG classification map of DEGs.** (A) COG classification diagram showing the DEGs between the sterile water and spore suspension treatments for variety #6; (B) COG classification diagram showing the DEGs between varieties #6 and #8 after spore suspension treatment.

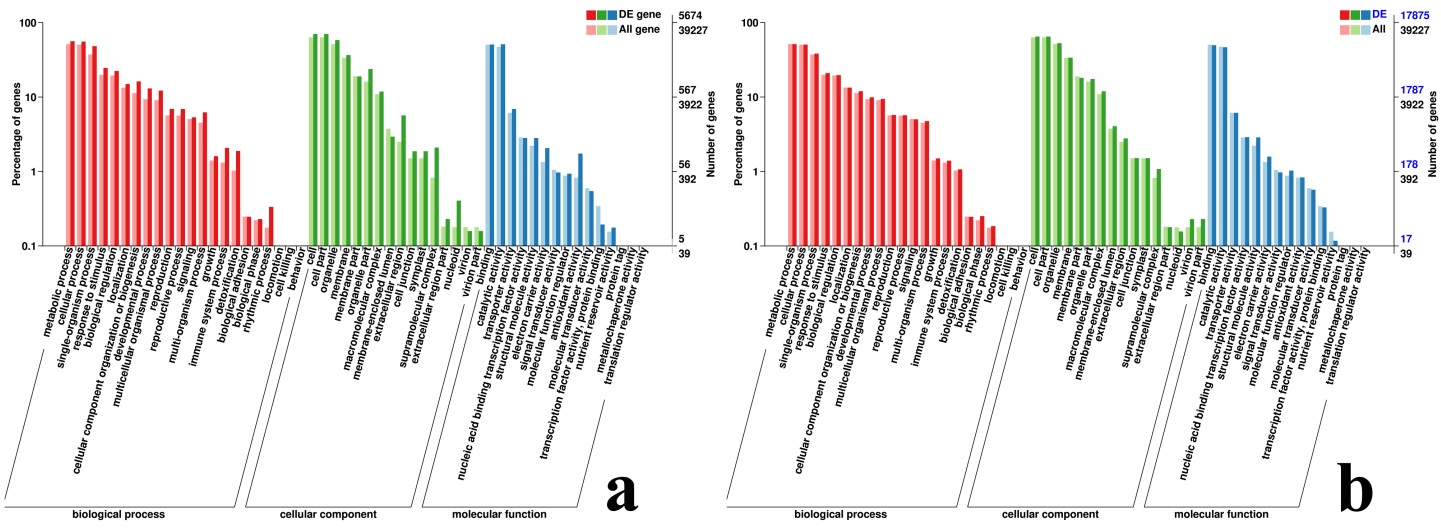

**Figure 5  GO classification map of DEGs.** (A) GO classification diagram showing the DEGs between the sterile water and spore suspension treatments for variety #6; (B) GO classification diagram showing the DEGs between varieties #6 and #8 after spore suspension treatment.

bamboo #8 inoculated with pathogens, most DEGs were enriched in the "phenylpropanoid biosynthesis" pathway (Fig. 6).

## RNA sequence analysis and annotation results between different varieties inoculated *A. phaeospermum* or sterile water

Firstly, the differentially expressed genes of #6 and #3, #6 and #8, #8 and #3 of the three varieties after inoculation with sterile water were compared (Table S3), and then after inoculation with *A. phaeospermum*, the comparison of #6 and #3, #6 and #8, and #8 and #3 excludes the difference after inoculation sterile water. When comparing DEGs from different *A. phaeospermum* inoculated varieties, 18401, 18496 and 4070 DEGs were identified between varieties #6 and #3, #6 and #8, and #8 and #3, respectively, including 4856, 5222, and 2380 upregulated genes and 13545, 13276, and 1690 downregulated genes (Table 2). In total, 1,401 DEGs were shared among #6 and #3, #6 and #8, and #8 and #3 (Fig. 3), of them, 279 DEGs were upregulated, and 528 DEGs were downregulated in all three comparison groups #6 and #3, #6 and #8, and #8 and #3 (Table S2). By comparing DEGs from the same varieties (including #3, #6, #8) under inoculation *A. phaeospermum* and sterile water treatment respectively, and all differentially expressed genes from # 6 and #3, #6 and #8, and #8 and #3 after inoculation *A. phaeospermum*, we found that the common DEGs were mainly related to energy metabolism (including photosynthesis and carbohydrate metabolism) and phytohormone biosynthesis, which indicated that energy metabolism and phytohormones played an important role in the resistance of *B. pervariabilis × D. grandis* to shoot blight.

After inoculation with *A. phaeospermum*, 1,467, 7,517 and 7,417 DEGs in #3 and # 6, #3 and #8, #6 and #8 were annotated into COG database respectively (Fig. S1). After inoculation with pathogens, the DEGs between #6 and #8 were annotated to the most COG

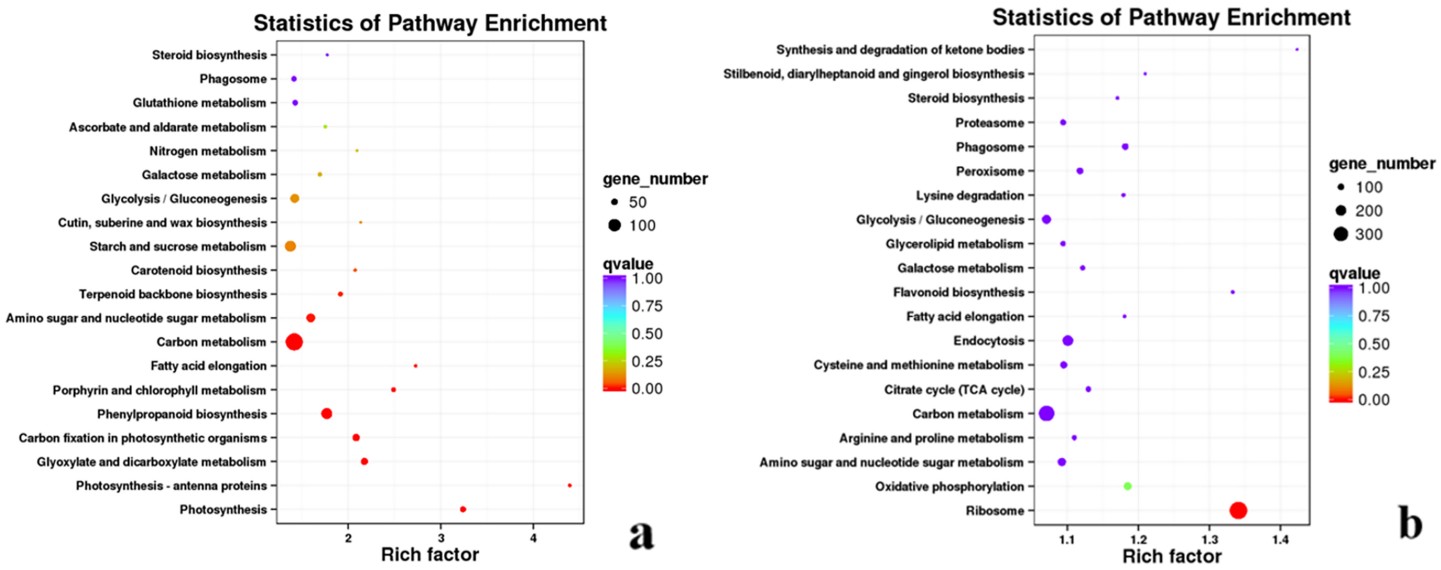

**Figure 6** **Summary of the KEGG enrichment maps of DEGs.** (A) KEGG classification diagram showing the DEGs between the sterile water and spore suspension treatments for variety #6; (B) KEGG classification diagram showing the DEGs between varieties #6 and #8 after spore suspension treatment.

database categories (7517 DEGs) across all comparison groups, among which most DEGs were annotated to "signal transduction mechanisms", "translation" and "replication, recombination and repair", with 1093, 1057 and 969, respectively (Fig. 4).

In the comparisons between #3 and #6, #3 and #8, and #6 and #8 of *B. pervariabilis* × *D. grandis*, there were 17,604, 3,658 and 17,875 DEGs, respectively, annotated to the cell component (Fig. S2), molecular function and biological process categories. In the analysis of hybrid bamboo #6 and #8 inoculated with pathogens, DEGs were enriched in "metabolic process", "cellular process" and "single organism process" in the biological process category. DEGs were mainly enriched in "cell", "cell part" and "organelle" in cell components. "Binding" and "catalytic activity" are molecular functions (Fig. 5).

After inoculation of varieties #3, #6 and #8 of *B. pervariabilis* × *D. grandis* with pathogens, 4557, 828 and 4689 DEGs were annotated to 127, 119 and 127 pathways for the comparisons #3 and #6, #3 and #8, and #6 and #8, respectively (Fig. S3). Across the three varieties, we found that the transcriptome data of the high-resistance varieties #6 and susceptible varieties #8 were enriched in the most KEGG pathways. These DEGs were mainly enriched in the "ribosome" pathway, followed by "carbon metabolism" and "endocytosis" (Fig. 6). These results suggest that energy metabolism- (including photosynthesis and carbohydrate metabolism) and transmembrane transport-related genes play important roles in the resistance of hybrid bamboo to shoot blight. Therefore, the downregulation of related DEGs may lead to a decrease in the disease resistance of hybrid bamboo, which is sensitive to *A. phaeospermum*.

**Table 3  Candidate genes screened from common DEGs.**

| Gene ID | Gene name | Annotation |
|---|---|---|
| PH01000030G0570 | *Tfbl* | transcription factor bHLH96-like |
| PH01000322G0910 | *Lrr-rlk2* | leucine-rich repeat receptor-like protein kinase IMK2-like |
| PH01000356G0220 | *Cer1l* | ECERIFERUM 1-like |
| PH01000743G0130 | *Gdsl-ll1* | GDSL-like lipase |
| PH01000967G0060 | *Prp4l* | proline-rich protein 4-like |
| PH01001075G0100 | *Scl45* | serine carboxypeptidase-like 45 |
| PH01001222G0140 | *Faah* | fatty acid amide hydrolase |
| PH01001261G0170 | *Klp* | kinesin-like protein FLA10 |
| PH01001610G0170 | *Plbp* | putative lipid-binding protein |
| PH01002098G0130 | *Xth28* | xyloglucan endotransglucosylase/hydrolase protein 28 |
| PH01002765G0010 | *C2agdcp* | C2 and GRAM domain-containing protein |
| PH01002901G0100 | *Laoh* | L-ascorbate oxidase homologue |
| PH01003226G0090 | *Atlp3* | auxin transporter-like protein 3 |
| PH01151705G0010 | *Gdsl-ll2* | GDSL-like Lipase |
| PH01000966G0470 | *Pod4l* | peroxidase 4-like |
| PH01000008G1560 | *Myb4l* | myb-related protein Myb4-like |
| PH01004943G0040 | *Pec53* | pectinesterase 53-like |
| newGene_56747 | *B-bwip1* | Bowman-Birk type wound-induced proteinase inhibitor WIP1 |
| newGene_63038 | *Egl20* | endoglucanase 20-like |
| newGene_67872 | *Pod16* | peroxidase 16 |
| newGene_84366 | *Cad5* | cinnamyl alcohol dehydrogenase 5 |

## Selection of candidate resistance genes for hybrid bamboo shoot blight and qRT-PCR verification

Based on GO and KEGG enrichment analysis, the DEGs related to plant resistance in the high enrichment pathway of DEGs were selected as candidate resistance genes. At the same time, the genes significantly differentially expressed in resistant and susceptible varieties and the same variety after inoculation with *A. phaeospermum* or sterile water were also selected as the object of disease resistance gene screening. Finally, through the comparison of existing studies, 21 candidate resistance genes were screened (Table 3). The qRT-PCR results of the candidate genes and five randomly selected genes, including P1rcsn, Pod2, Hltfah1, Clbd, and Pod4 (Table 4), as well as the results for the reference gene GAPDH, showed a single peak, and the relative expression level was consistent with the transcriptome data (Fig. 7).

## DISCUSSION

Transcriptome sequencing technology is an important means to explore genes and a powerful tool to identify the interaction network between plants and pathogens. The xylogenesis formation of Moso bamboo shoot was studied by transcriptome sequencing (*Zhang et al., 2018*), the transcriptome of *Dendrocalamus latiflorus* Munro was sequenced and de novo analyzed using Illumina platform, and many key candidate genes related to

**Table 4** Five genes that were randomly selected to verify the expression level.

| Gene ID | Gene name | Annotation |
| --- | --- | --- |
| PH01087490G0010 | *PIrcsn* | photosystem I reaction centre subunit N |
| PH01000445G0570 | *Podbd2* | peroxidase |
| PH01000974G0460 | *Hltfah1* | histone-like transcription factor and archaeal histone |
| PH01000040G1380 | *Clbd* | calreticulin-like |
| PH01000445G0440 | *Podbd4* | peroxidase |

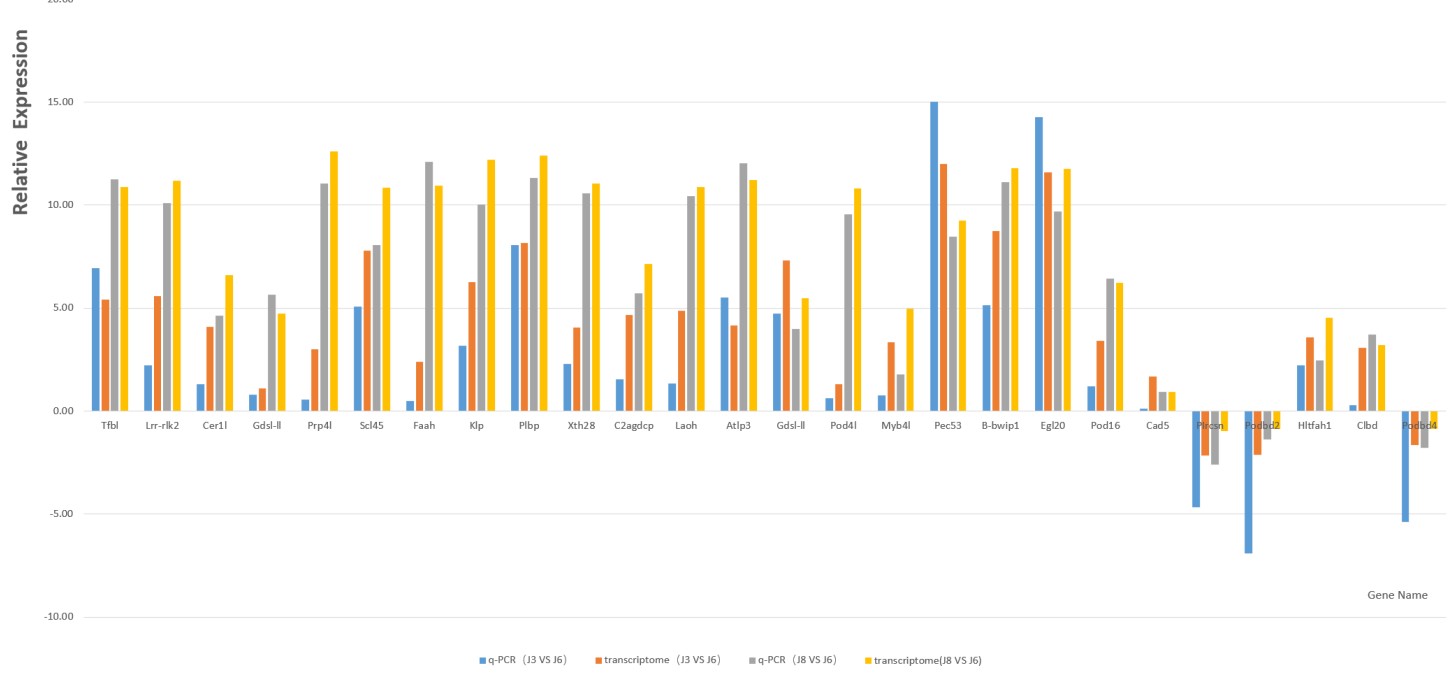

**Figure 7** Column map of J8, J6 and J3 gene expression measured by qRT-PCR.

the growth and development of *Dendrocalamus latiflorus* were found (*Liu et al., 2012*). In this study, transcriptome sequencing technology was used to study the reaction of three varieties of hybrid bamboo inoculated with water or pathogens. Transcriptome sequencing analysis showed that under six different conditions (S3, J3, S6, J6, S8, J8), many DEGs may be potential resistance genes in *B. pervariabilis × D. grandis*. Combined with gene functional annotation, candidate disease resistance genes were determined; these genes may be the key to determining the disease resistance mechanism of *B. pervariabilis × D. grandis* regarding shoot blight. In addition, 21 candidate genes and five randomly selected genes were verified by qRT-PCR, and the results showed that these genes had 100% similarity with the transcriptome sequencing results, confirming their authenticity.

Plant resistance to pathogens involves several different mechanisms. On the one hand, some structural and chemical components of plants have disease resistance functions, such as cutin, wax, embolus, lignin and special stomatal structures of the cell wall; on the other

hand, plants show disease resistance reactions through the regulation of resistance genes when infected by pathogens (*Dhaliwal & Uchimiya, 1999*).

The cell wall is the first barrier against pathogen invasion, and pectin is the main component of the primary cell wall (*Caffall & Mohnen, 2009*). Pectin methylesterase (PME), also known as pectinesterase, is one of the main enzymes that modifies pectin. This enzyme plays an important role in the synthesis of the cell wall and the regulation of elasticity and permeability. Studies have shown that PME activity in Arabidopsis thaliana is significantly enhanced under pathogen infection (*Bethke et al., 2013*). PME not only regulates the growth and development of the plant cell wall but also participates in the pathogenic process of pathogens. PME participates in the demethylation process of pectin, promotes the further decomposition of pectin, and reduces the strength and hardness of the cell wall. In *A. thaliana*, *AtPME3* was upregulated after infection with *Botrytis cinerea* and *Pectobacterium carotovorum*, and the expression of *AtPME3* was considered a susceptibility factor in *A. thaliana* (*Raiola et al., 2011*). However, oligogalactic acid (OG) is an elicitor-active substance, and its degraded products can be recognized by downstream kinase receptors and further activate downstream defence responses, thereby improving plant disease resistance (*Villegas et al., 2016*). In this study, the expression level of the PH01004943G0040 gene (annotated as pectinesterase 53-like) was significantly increased in the high-resistance varieties #6 after inoculation, indicating that this gene may be closely related to the resistance of *B. pervariabilis × D. grandis* to shoot blight. However, whether the downstream defence response was activated by producing OG in *B. pervariabilis × D. grandis* needs further experimental confirmation. There was almost no expression of pectinesterase 53-like after inoculation with *A. phaeospermum* in varieties #3 and #8, indicating that the expression level of pectinesterase 53-like increased with increasing resistance in hybrid bamboo infected with *A. phaeospermum*, which further indicated that *pectinesterase 53-like* might be involved in the resistance response of hybrid bamboo to *A. phaeospermum*.

In plant-pathogen interactions, the increase in lignin is an important mechanism for plants to resist pathogens and enhance disease resistance, as lignin is involved in a varieties of physiological and biochemical reactions in plants. Cinnamyl alcohol dehydrogenase (CAD) was isolated from tobacco stems for the first time in 1992 (*Knight, Halpin & Schuch, 1992*); this enzyme was previously found to be involved in lignin synthesis and played a key role in lignin synthesis (*Wyrambik & Grisebach, 1975*). The number of CAD family members in different plants varies, and nine, 12, 16, 15, 14 and 17 CAD family members have been reported in Arabidopsis, rice, Populus trichocarpa, Populus hybrida, sorghum and alfalfa (*Kim et al., 2004*; *Tobias & Chow, 2005*; *Kim et al., 2004*; *Shi et al., 2010*; *Barakat et al., 2009*; *Saballos et al., 2009*). Many factors in the natural environment, such as hormones, mechanical damage, fungi and mineral elements, can induce CAD to participate in the plant lignification defence process (*Kim et al., 2007*), which reduces the adverse effects of external factors on plants and plays an important role in plant growth and development. *CAD-C* and *CAD-D*, the main genes for lignin biosynthesis in *A. thaliana*, were found to be important genes for defence against strong and weak varieties of *Pseudomonas syringae pv. tomato*. In this study, new_gene_84366 (annotated as cinnamyl

alcohol dehydrogenase 5) was significantly upregulated after pathogen inoculation of the hyperactive cultivar #6, indicating that the gene may be closely related to the resistance of *B. pervariabilis* × *D. grandis* to shoot blight.

After plants are infected with pathogens, the metabolic balance between the production and elimination of reactive oxygen varieties in the body is damaged. Excess reactive oxygen varieties can trigger and aggravate membrane lipid peroxidation. The cell structure is destroyed, and the metabolic system is disordered, which will lead to plant cell death in serious cases. POD is the key enzyme in the protective enzyme system that is responsible for scavenging reactive oxygen varieties in plants. The increase in POD activity indicates that the ability to scavenge free radicals is enhanced and that membrane lipid peroxidation is reduced, thereby maintaining the normal metabolism of cells. In addition, POD plays an important role in the synthesis of antibacterial substances such as lignin, phenols and phytoalexin (*Hwang et al., 1997*; *Yubedee, 2013*; *Green, 1975*; *Cao et al., 2013*). In the transcriptome data for the different treatments of *B. pervariabilis* ×*D. grandis*, we found that the expression levels of PH01000966G0470 and new gene_67872 (annotated as peroxidase-like) increased after inoculation in #3 and #6 hybrid bamboo, especially in the highly resistant varieties #6, indicating that the POD synthesis reaction was activated during the resistance of *B. pervariabilis* × *D. grandis* to shoot blight, and this reaction was more significant in resistant varieties. This gene may be closely related to *B. pervariabilis* × *D. grandis* resistance to shoot blight.

Plant GDSL lipase is a large gene family and plays an important biological role in plant growth, morphogenesis, lipid metabolism and the defence response (*Kondou et al., 2008*; *Takahashi et al., 2009*). The expression of GDSL lipase can change in response to biotic and abiotic stresses. The expression of GDSL lipase in plants can be induced by hormones such as pathogens, salicylic acid, ethylene, jasmonic acid and abiotic stress factors, indicating that this gene may be involved in plant disease resistance and stress responses (*Oh et al., 2005*; *Kram et al., 2008*; *Lee & Cho, 2003*; *Kim et al., 2013*). GDSL lipase can regulate or directly destroy the integrity of fungal spores through signal transduction, limiting the growth and reproduction of pathogens in infected areas (*Oh et al., 2005*; *Kwon et al., 2009*). PH01151705G0010 and PH01000743G0130 (annotated as GDSL-like lipase) were detected in the three varieties, and their expression levels increased with increasing disease resistance, especially in highly resistant varieties. Their expression levels increased significantly after inoculation, indicating that GDSL lipases may be involved in the disease resistance process of *B. pervariabilis* ×*D. grandis* against shoot blight, and this gene family may be the key resistance genes in *B. pervariabilis* ×*D. grandis*.

In addition, in the transcriptome data analysis, we also found a gene encoding MYB protein in the highly resistant varieties #6. The MYB family is the most abundant transcription factor family in plants and can regulate plant development, metabolism, physiological rhythm regulation and key factors for biological and abiotic stress responses (*Dubos et al., 2010a*; *Hosoda et al., 2002*; *Dubos et al., 2010b*). Studies have shown that many MYB genes are involved in the plant response to stress. The molecular expression characteristics of 60 MYB genes in wheat under high salt, ABA and PEG stress showed that the expression of *TaMYB48* with a MYB domain was first upregulated and then

downregulated (*Zhang et al., 2012*). Hiroshi (*Abe et al., 1997*) found that a MYB protein (ATMYB2) can activate the expression of the rd22 gene and ABA gene in Arabidopsis thaliana, thereby improving plant resistance. In this study, the expression level of the PH01000008G1560 gene (annotated as myb-related protein Myb4-like) was significantly higher in the highly resistant varieties #6 than in the susceptible varieties #3 after inoculation, indicating that the PH01000008G1560 gene may be closely related to the resistance of *B. pervariabilis ×D. grandis* to shoot blight. At the same time, we found that the expression level of Myb4-like was lower after inoculation with *A. phaeospermum* than after treatment with sterile water in the susceptible varieties #8 and the moderately resistant varieties #3, which may be due to the inhibition of Myb4-like expression by ABA accumulation. The specific result needs further experimental confirmation.

In this study, many DEGs involved in lignin and phytoalexin synthesis, redox reactions, photosynthesis and signal transduction were found to be potentially important disease resistance genes. The results of this study provide a basis for the further study of resistance genes and related molecular functions in *B. pervariabilis × D. grandis* related to shoot blight caused by *A. phaeospermum* and provide a basis for further study of its regulatory mechanism. The transcriptome data of *B. pervariabilis × D. grandis* were uploaded for the first time and laid the foundation for molecular breeding of *B. pervariabilis × D. grandis* in the future.

## CONCLUSIONS

*The B. pervariabilis ×D. grandis* transcriptome was first sequenced and uploaded to the public database. *B. pervariabilis ×D. grandis* with different resistance varieties showed different levels of disease resistance after being infected with *A. phaeospermum*. #3, #6 and #8 showed disease resistance, high resistance and susceptibility, respectively. After inoculation with *A. phaeospermum* in #6 resistant varieties, many genes related to disease resistance were also significantly expressed, especially those related to transcription, energy metabolism and cell wall composition synthesis. The results showed that the expression of 21 genes such as *Gdsl-ll*, *Myb4*, *Pec53*, *Cad5* and *Pod16* was significantly increased after inoculation with #6 varieties. These genes may be involved in the resistance response of *B. pervariabilis ×D. grandis* shoot blight caused by *A. phaeospermum*, and they are important potential resistance genes. The specific functions of these genes in the resistance process of *B. pervariabilis × D. grandis* need further experimental verification.

### Funding
This research was supported by the National Natural Science Foundation of China (grant number 32171795) and the General Program of Science and Technology Department of Sichuan (grant number 2020YJ0400). The funders had no role in study design, data collection and analysis, decision to publish, or preparation of the manuscript.

## Grant Disclosures

The following grant information was disclosed by the authors:

The National Natural Science Foundation of China: 32171795.

The General Program of Science and Technology Department of Sichuan: 2020YJ0400.

## Competing Interests

The authors declare there are no competing interests.

## Author Contributions

- Fengying Luo, Xinmei Fang and Shujiang Li conceived and designed the experiments, performed the experiments, prepared figures and/or tables, authored or reviewed drafts of the paper, and approved the final draft.
- Han Liu, Tianhui Zhu and Shan Han performed the experiments, authored or reviewed drafts of the paper, and approved the final draft.
- Qi Peng performed the experiments, analyzed the data, authored or reviewed drafts of the paper, and approved the final draft.

## Data Availability

The raw date is available at BioProject: PRJNA545783; BioSample: SAMN11928018; TSA: GJJA00000000.

## Supplemental Information

Supplemental information for this article can be found online at http://dx.doi.org/10.7717/peerj.12301#supplemental-information.

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
