# Peer review of "Differential transcriptome analysis and identification of genes related to resistance to blight in three varieties of Bambusa pervariabilis × Dendrocalamopsis grandis"

_PeerJ, doi:10.7717/peerj.12301_

## Round 0.1 · original submission · Major Revisions

Dear Dr. Li and co-authors,


As you can see, our reviewers found that your study was important,
however they provided several comments and suggestions to strengthen
your manuscript.

Reviewer 1 had some essential comments regarding the expression
differences by the genetic background and the identification of DEGs
significantly related to disease resistance. Reviewer 2 also provided
valuable suggestions and comments, such as needs for detail explanation of your comparison with various materials and/or treatments and method description with the functional databases, like GO, KEGG, and COG. Reviewer 3 pointed out the weak points in the main text, especially related to reproducibility and transparency. For example, you should explain all the parameters/information of the tools used in this study, like HISAT2. It is important to share all your transcript data with
public repositories, such as NCBI TSA.

I fully agree with their comments and suggestions. I would like to ask
you to address or to respond with reasons not to follow the suggestion
made by these reviewers.


Best regards,
Atsushi Fukushima

Reviewer 1 ·

Basic reporting

The Ms was well writeen in English, technically correct text, the situation of bamboo and the research progress were described in detail.

Experimental design

By sequencing the transcriptome of three hybrid offspring before and after infection, the experimental design is basically reasonable. And I think that the phenotypic changes before and after infection must be added, especially the differences in disease resistance, so as to lay a foundation for transcriptome sequencing and differential gene screening.

Validity of the findings

The sequencing results was analyzed and differentially expressed genes were screened, but which of these DEGs are related to disease resistance? In addition, how can the authors exclude the gene expression differences caused by the genetic background differences of the three hybrid offspring 3,6,8#? Therefore, gene expression differences caused by genetic background differences should be excluded in differential gene analysis.

Reviewer 2 ·

Basic reporting

In this manuscript, comparative transcriptomics using RNA-seq data were applied to analyze different expressed genes (DEGs) in three different species of B. pervariabilis × D. grandis with variable resistance and in the same specie under different treatments. The functional genes identified here are important genetic resources to improve the resistance of B. pervariabilis × D. grandis to shoot blight.

Experimental design

Generally, the A. phaeospermum inoculated materials were compared with their respective water controls, and then compared with each other to identify DEGs. Why did you perform two kinds of transcriptome comparison to identify DEGs? One is between A. phaeospermum infection and sterile water inoculation in the same species. The other is in different A. phaeospermum inoculated varieties. You can discuss this.

Validity of the findings

No comment

Additional comments

For introduction
Too many descriptions about the infection symptoms of A. phaeospermum in bamboos. Author cited many research on host-pathogen interaction and application of transcriptomics which were performed on other crops. No related studies on hybrid bamboos?

For Materials and methods
1. Line113, how were spores cultured?
2. Please provide the link or accession number for your transcriptome raw data in the manuscript.
3. Differentially expressed gene analysis section. I think it is better to describe in detail how are compared among various materials or various treatments, and write clearly what J3, J6, J8, S3, S6, and S8 stand for. This would make reader to understand the results easier.
4. Line 154, The authors do not adequately state their procedures for qRT-PCR. Please add the appropriate details or add the literature.

For results
1. Line 167. “Comparative analysis of transcriptomic expression data showed that 4387, 6316 and 4159 DEGs were obtained from the #3, #6 and #8 B. pervariabilis × D. grandis varieties inoculated with A. phaeospermum or sterile water, respectively”. It's not easy to understand what you mean without reference to Table 2. I suggest the following changes: “Comparative analysis of transcriptomic expression data was determined between A. phaeospermum infection and sterile water inoculation in the same species. The results showed that 4387, 6316 and 4159 DEGs were obtained in the #3, #6 and #8 varieties, respectively.”
2. Line 175-178. “In the #3, #6 and #8 B. pervariabilis × D. grandis varieties after inoculation, a comparative analysis of differences in transcriptomic expression between varieties #6 and #3, #6 and #8, and #8 and #3 found 22053, 22464, and 4638 DEGs, respectively, including 7295, 7936, and 1778 upregulated genes and 14758, 14528, and 2860 downregulated genes.” I suggested reword this long sentence to “When comparing DEGs from different A. phaeospermum inoculated varieties, 22053, 22464, and 4638 DEGs were identified between varieties #6 and #3, #6 and #8, and #8 and #3, respectively, including 7295, 7936, and 1778 upregulated genes and 14758, 14528, and 2860 downregulated genes.”
3. Line 179. “In total, 2154 DEGs were shared among the three varieties (Fig. 2); of them, 23 DEGs were upregulated in the three species, and 143 DEGs were downregulated (Table S2)”. Were all 23 genes up-regulated and 143 genes down-regulated in all three varieties? Please provide the expression of each gene in each treatment in Table S2.
4. Line180, ‘By comparing these DEGs, ------’, these DEGs referred what?
5. Line 186-188. ‘In the COG database, 1513, 2776, and 1421 and 1467, 7517, and 7417 DEGs were found for #3, #6 and #8 B. pervariabilis × D. grandis after inoculation with A. phaeospermum or sterile water, respectively.’ this sentence was more confused. Which genes were used for COG annotation, and how many genes were annotated in COG database?
6. Line 189, I couldn't understand what was said in Fig. S1 due to its notes vague in the figure. What does a, b, c, d, e and f stand for?
7. Line 238-241. “Candidate genes were screened from the group of common differentially expressed genes from multiple comparison groups. Based on the enrichment of upregulated DEGs in the GO and KEGG databases and combined with functional annotation information, 21 candidate genes for disease resistance were selected”. Why screening the candidate genes from the common DEGs? How selected 21 candidate genes based on the enrichment of upregulated DEGs in the GO and KEGG databases? Please discussion this.
For discussion
1. The discussion mainly discussed the genes listed in Table 3. You can add discuss about the materials and methods used in this research.
2. Line 257, please give the full name of PEG.

Reviewer 3 ·

Basic reporting

The authors sequenced the transcriptomes of three varieties of hybrid bamboos with different degrees of pathogen resistance, and compared differentially expressed genes before and after inoculation, and between different varieties.

Experimental design

Generally, the experimental designs are feasible. But the processes should be clearer. For example, the tools used to identify DEGs.

Validity of the findings

The findings are reasonable.

Additional comments

Line 24 specie?
Lines 28-30 This sentence is unclear.
Line 31 another expression of the former sentence
Line 41 what’s the difference between species in title and varieties here?
Line 75 Liberibacter asiaticus, should be italic?
Line 109 use species or variety should be uniform across the paper
Line 134 HISAT2, version and parameters should be provided
Line 136-138 Is this the link to the reference genome? you can cite their reports or the database. Can hisat2 be used to align short reads to transcriptome assembly? Moreover, you should provide the Transcriptome Shotgun Assembly (TSA) accession.
Line 141 generally, FPKM is superior than RPKM and was widely used at present. What is the tool you used to identify differentially expressed genes?
Line 170 species? Or varieties?
Lines 173-174 you said 394 DEGs were shared, but why only 74 genes were un- and downregulated? How about the 320 shared genes?
Lines 175-178 you should think about the constitute differences between these varieties.
Line 179-180 similar to lines 173-174
Lines 185-235 should be separated and placed after the suitable results of differential gene analysis.
Table 2 what is J and S? should be included in the table legend.

---

## Round 0.2 · accepted · Accept

Dear authors,

Thank you for revising. You have answered and responded well.

Best regards

Reviewer 1 ·

Basic reporting

No other comments

Experimental design

No other comments

Validity of the findings

Through bacterial inoculation, transcriptome sequencing and differential gene analysis, the MS finally screened a series of key DEGs, and analyzed the expression of five DEGs genes. The MS has been modified according to advices of the reviewer, so the MS can be accepted.

Additional comments

No other comments

Reviewer 3 ·

Basic reporting

no comment

Experimental design

no comment

Validity of the findings

no comment

Additional comments

All points have been well addressed.